# Excavating hidden adsorption sites in metal-organic frameworks using rational defect engineering

Sanggyu Chong [1], Günther Thiele[2] & Jihan Kim[1]

Metal–organic frameworks are known to contain defects within their crystalline structures. Successful engineering of these defects can lead to modifications in material properties that can potentially improve the performance of many existing frameworks. Herein, we report the high-throughput computational screening of a large experimental metal–organic framework database to identify 13 frameworks that show significantly improved methane storage capacities with linker vacancy defects. The candidates are first identified by focusing on structures with methane-inaccessible pores blocked away from the main adsorption channels. Then, organic linkers of the candidate structures are judiciously replaced with appropriate modulators to emulate the presence of linker vacancies, resulting in the integration and utilization of the previously inaccessible pores. Grand canonical Monte Carlo simulations of defective candidate frameworks show significant enhancements in methane storage capacities, highlighting that rational defect engineering can be an effective method to significantly improve the performance of the existing metal–organic frameworks.

[1] Department of Chemical and Biomolecular Engineering, Korea Advanced Institute of Science and Technology (KAIST), Daejeon 34141, South Korea.
[2] Department of Chemistry, University of California Berkeley, Berkeley, CA 94720, USA. Correspondence and requests for materials should be addressed to J.K. (email: jihankim@kaist.ac.kr)

Metal–organic frameworks (MOFs) are crystalline porous materials composed of organic linkers and metal ions or clusters connected by coordinative bonds, often resulting in ultrahigh surface area and porosity. This simple motif of combining different metal nodes and organic linkers has led to the discovery of nearly 70,000 experimentally synthesized MOFs[1,2] and more than 100,000 hypothetical MOFs[3] that are unique in their chemical composition and topology. High chemical tunability over the organic linkers and metal clusters make MOFs promising candidates for a wide variety of applications, ranging from catalysis[4] to gas separation[5] and storage of alternative energy sources such as methane and hydrogen[6,7].

Although a large number of MOFs have been experimentally synthesized to date, researchers are generally interested in a select few MOFs (e.g., HKUST-1, MOF-74, MIL-53, UiO-66, and ZIF-8) renowned for exhibiting exceptional attributes well suited for the development of important, industrial applications. The majority of other synthesized MOFs are mostly neglected after the development of another material that surpasses their performance in the targeted application field. With such an enormous set of experimental MOFs being overlooked, it is worthwhile to investigate the ways in which these preexisting MOFs can be recycled and modified to show pronounced improvements in their material properties, perhaps rendering them to be promising for applications different from what was initially intended.

To further improve the performance of existing MOFs, one must ask the question of whether or not a given framework is performing at its fullest capacity. One possibility that may limit a MOF from reaching its maximum potential would be the presence of unexploited volumes within the framework that are inaccessible toward the gas molecules of interest. This notion of inaccessibility is not absolute and is highly dependent on the effective pore aperture of the adsorbent and size/shape of the adsorbate. In the case of zeolites, several studies have confirmed the existence of inaccessible pores in some structures (e.g., LTA, FAU, and DDR) and how it affects their gas adsorption capacities[8–10]. MOFs tend to be more flexible than zeolites, making it difficult to ascertain inaccessibility from a theoretical point of view[11–13]. Nonetheless, several studies have established the presence of inaccessible pores in several MOFs and emphasized the crucial importance of accurately accounting for these pores in predicting the performance of MOFs[14–16]. Given thousands of experimentally synthesized MOFs, we hypothesize that many of them will also exhibit inaccessible pores whose presence has been neglected for the most part. These inaccessible pores can be viewed as potential interaction sites yet to be exploited, and as such, the goal of this work is to develop a methodology in which these sites can be freshly excavated to significantly improve the adsorption properties of MOFs.

One possible way to achieve this is via defect engineering. By exercising control over the defects present within MOFs, desirable chemical properties can be intensified for enhanced adsorption properties. Although defects within zeolites have been extensively studied, defects within MOFs have not been given much attention until recently[17]. Understanding and even exercising control over the formation and distribution of defects within MOFs will be crucial for new impactful applications of these nanoporous materials[18–20]. Although several different types of defects can be present within MOFs[21–25], this study primarily focuses on zero-dimensional linker vacancy defects that are more easily controlled.

The presence of linker vacancies within MOFs can play a crucial role in their resulting chemical behavior. In the symbolic case of UiO-66, researchers have attributed the unusual hydrophilicity and enhanced catalytic activity of the framework to the presence of linker vacancies that ultimately result in additional unsaturated metal sites[26–29]. Linker vacancy defects within MOFs also alter their physical attributes, generally leading to a decrease in surface area but an increase in pore volume. However, these changes are often negligible in the scopes of gas adsorption. A recent simulation study showed that in IRMOF-1, periodically distributed linker vacancies of up to 20% did not significantly alter the pore size distribution or the resulting argon adsorption profile[30]. Moreover, Barin et al.[31] synthesized NU-125 and HKUST-1 with fragmented linkers to create vacancy defects, with the goal of enhancing the pore volumes and surface areas of these materials. Nevertheless, gas (i.e., $CO_2$, $CH_4$, and $H_2$) uptakes decreased for HKUST-1 and increased only slightly for NU-125, with the working capacities nearly being the same in both materials.

On the other hand, Jiang et al.[32] successfully showed a $CO_2$ uptake enhancement of around 60% in USTC-253 at 1 atm and $T = 273$ K by introducing linker vacancy defects to create new unsaturated metal sites. However, these defects could not secure adequate pore volume in USTC-253 for $CO_2$ uptake enhancement to persist in the high-pressure regime. Wu et al.[33] demonstrated that with controlled introduction of linker vacancy defects in UiO-66, the gas adsorption capacities in the high-pressure regime can improve significantly (i.e., up to 30% for $CH_4$ at 60 bar and 50% for $CO_2$ at 35 bar, both at 300 K). However, this enhancement in gas adsorption was attributed to the presence of numerous mesopores, which can no longer be considered as zero-dimensional point defects. All in all, previous studies were unable to show a clear trend of significant adsorption enhancement throughout all pressure regimes for any of the MOFs with zero-dimensional linker vacancies alone.

In this study, we combine two concepts of "linker vacancy" and "inaccessibility" to pave the way to excavate the inaccessible pores and improve the gas adsorption capacities of the existing MOFs. If the linkers barricading the inaccessible regions could be altered or removed, previously inaccessible pores can be merged with the main adsorption channels for significant expansion of both the

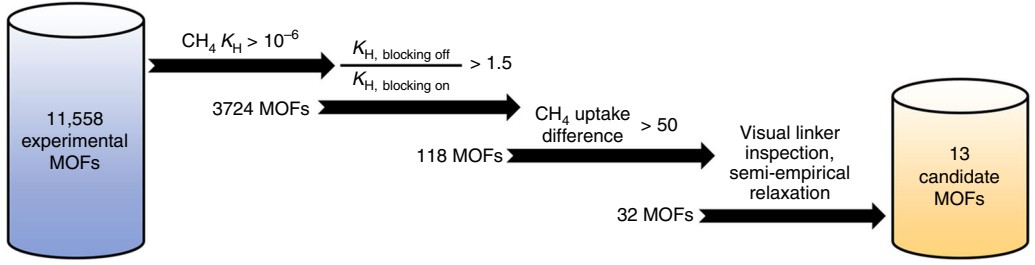

**Fig. 1** Overall screening process adopted in this research. Starting from the extended data set of computation-ready experimental MOFs, a refined number of candidate frameworks under each stage of the screening process is presented. Henry coefficient ($K_H$) is measured in mol kg$^{-1}$ Pa$^{-1}$ and methane uptake is measured in (v STP/v)

surface area and pore volume. We conduct a high-throughput computational screening on a large number of experimental MOFs to identify MOFs with significantly large enhancement of methane uptake with < 10% of linker vacancy defects, suggesting the feasibility of using rational defect engineering to improve gas adsorption performance.

## Results

**Screening of CoRE MOF database.** It is not a priori clear which MOFs can show significant uptake enhancements upon introducing linker vacancy defects. As such, we have devised a method that allows us to judiciously search through the MOF materials space to accurately and efficiently find the appropriate candidates (Fig. 1). A large-scale computational screening was conducted on an extended version of computation-ready experimental (CoRE) MOF data set[1] prior to application of the porosity filter, and also the DFT-minimized MOF data set[34], for a total of 11,558 structures with a unique Cambridge Structural Database (CSD) reference codes. There have been many other screening work using the CoRE MOF structures in the past[35–39] but to the best of our knowledge, our study is the first to focus on exploiting the potential gains from modifying the original CoRE MOF structures.

Methane was chosen as the gas molecule of interest to effectively ignore the contributions from newly created unsaturated metal sites (given the lack of dipole and quadrupole moments possessed by methane) and to solely observe the effects of surface area and pore volume expansion. Also, it has been previously shown that generic force fields for methane generally lead to accurate predictions of the adsorption properties of MOFs, making our results more reliable[3, 35]. $CH_4$ Henry coefficients ($K_H$) were calculated for each MOF twice, once with the blocking feature on and once with the feature off, to effectively screen for the presence of inaccessible pores. As our first screening criterion, structures with $CH_4$ $K_H < 10^{-6}$ mol kg$^{-1}$ Pa$^{-1}$ at $T = 298$ K were disregarded to guarantee ample porosity toward methane in all of our candidates. This criterion narrowed down the number of MOFs to 3724. Then, we subsequently screened for candidates with a $K_H$ ratio > 1.5, where $K_H$ ratio is defined as follows:

$$K_H \text{ ratio} = \frac{K_{H,\text{ blocking off}}}{K_{H,\text{ blocking on}}} \quad (1)$$

This was done to guarantee significant enough changes in the adsorption properties with the consideration of an inaccessibility phenomenon. A total of 3481 structures showed no difference in $CH_4$ $K_H$ with and without blocking. The distribution of the remaining 243 structures with a $K_H$ ratio > 1 is shown in Fig. 2, and 118 MOFs were found to exhibit a $CH_4$ $K_H$ ratio of higher than 1.5.

With the refined pool of candidates, $CH_4$ grand canonical Monte Carlo (GCMC) simulations were performed twice at $T = 298$ K to calculate the adsorption isotherms of the candidate MOFs, once with and once without the blocking feature. The simulation results are presented in Fig. 3. A total of 32 MOFs with an absolute uptake difference between blocking on and off of higher than 50 (v STP/v) at 65 bar (relevant as methane storage conditions for ANG and a good enough proxy for high-pressure conditions) were identified and taken for further considerations. This degree of difference in the high-pressure regime shows that there can exist a significant volume of inaccessible pores, which could potentially be opened up with linker vacancy defects for additional adsorption of methane. Yet, 50 (v STP/v) is an arbitrarily chosen cutoff value, and some remaining MOFs that did not make this cutoff can still experience methane uptake

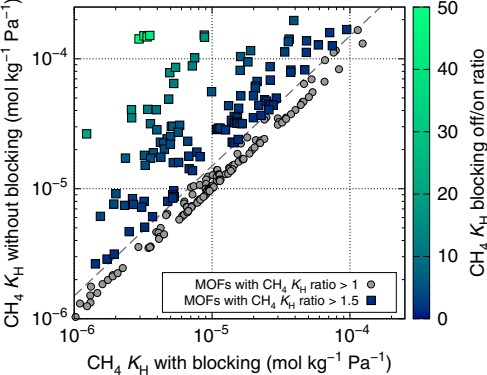

**Fig. 2** Methane Henry coefficients of each candidate MOF with and without blocking. This logarithmic graph shows methane $K_H$ ratios of each MOF with and without the blocking feature at $T = 298$ K for MOFs with a $K_H$ ratio > 1. The gray dashed line has been drawn where $K_H$ ratio = 1.5. A total of 118 MOFs were found to have a $CH_4$ $K_H$ ratio of higher than 1.5

enhancement from linker vacancies. As such, we have included a list of 50 additional MOFs in Supplementary Table 1.

At this point, the remaining candidate MOFs were manually inspected for their coordination environment and linker vacancy creation possibilities, and the inspection results are presented in Supplementary Table 2. By performing the visual inspection, we aimed to guarantee the feasibility of creating linker vacancy defects to the best of our knowledge. First, we actively opted for MOFs containing linkers coordinating with carboxylate or azole end groups only, on which a sufficient amount of previous research effort has been spent on how their linker vacancies may be expressed[28, 40, 41]. In addition, we also aimed to find and discard MOFs containing linkers with more than three binding groups, where we reasoned that exposure of too many metal sites cannot be stabilized well within a MOF structure. During the visual inspections, two porphyrin MOFs (DAPBIH and DAN-ZOJ)[42] were omitted for the erroneous removal of porphyrin during the experimental MOF database construction. Six MOFs (XAL series)[43] comprised of highly complex ligands were omitted for failing our inspection criteria of linker denticity. One MOF (OYUJUO)[44] was found to be identical to another MOF (AXUBOL)[45] and was thus eliminated for redundancy, leaving us with 23 MOFs.

Before moving forward, the 23 candidate CoRE MOF structures were relaxed using MOPAC and new $CH_4$ GCMC simulations were performed on the candidates to test and see whether the inaccessible regions would persist upon energy minimization of the frameworks. In the process of MOPAC relaxation, it was found that the candidate pool contained several anionic MOFs whose countercations were missing in the structure files. The cations were then manually added for the correct representation of these MOFs, and the cation insertion procedure is briefed case by case in Supplementary Discussion 1. Two lanthanide MOFs (XOMJOY and XOMJUE)[46] were found to be incompatible with the MOPAC program in the process and were unfortunately removed from our candidate pool. Three other MOFs (EHUFAP, EZOFEF, and PEYVEV)[47–49] were ruled out for showing a significant structural collapse or linker detachment with MOPAC energy minimization, and five more (SAKNOJ, VET series)[50, 51] were unfortunately omitted for being virtually incompatible with MOPAC relaxation due to an undesirably large number of atoms in the unit cell. As for the remaining 13 candidate MOFs, an inaccessibility phenomenon persisted after energy minimization with MOPAC and was

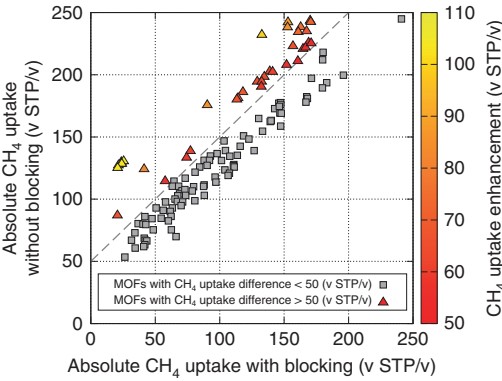

**Fig. 3** Absolute methane uptake of each candidate MOF with and without blocking. GCMC simulation results with and without blocking at $P = 65$ bar and $T = 298$ K for the 118 MOFs with a $K_H$ ratio higher than 1.5 are presented. A dashed line is drawn where the uptake difference with and without blocking is equal to 50 (v STP/v). Only 32 MOFs were found to have uptake enhancements at 65 bar that surpasses our criterion

confirmed to be ready for the introduction of linker vacancy defects.

**Creation of linker vacancies in candidate MOFs**. Prior to introducing the linker vacancy defects, unit cells of the relaxed candidate MOFs were expanded accordingly, so that with a single linker vacancy, the total proportion of linker vacancies within the framework would be kept to $< 1/12$. This was done to maintain the fraction of linker vacancies lower than what has been commonly reported for UiO-66[22, 23, 33], which, given the high-coordination environment of UiO-66, may serve as a crude upper limit of an attainable defect proportion in MOFs. It should be noted, however, that even higher linker defect rates can be experimentally observed in MOFs[52], and hence, it may also be possible for the candidate MOFs to be engineered to withstand higher defect rates than those considered here. After appropriate unit cell expansions, linker vacancy defects were introduced by removing a single linker from the unit cell and substituting it with the appropriate modulators or solvent molecules. Sample defect introduction schemes are shown in Fig. 4. In creating linker vacancies coordinated with modulators, formates were used to replace the carboxylate end groups[28, 40] and appropriate azolates were used to replace the azole-containing linkers[41]. In inserting the monodentate modulators, the original positions of the overlapping atoms (e.g., O = C–O of a carboxylate-containing linker and formate modulators) were left undisturbed. As for the case of linker vacancies with water and hydroxides, we first incorporated enough hydroxides to retain the charge balance at each of the metal clusters involved, and the remaining metal sites were populated with water. When necessary, a *trans* configuration between water and hydroxide was used across metal clusters, as it was reported to be lower in energy than the *cis* configuration[40].

By expressing the linker vacancies as described, we adopted a conservative approach of expressing defect frameworks for gas adsorption. Experimentally, modulators and solvent molecules can break free and evaporate to expose new metal sites upon activation of MOF for gas adsorption. However, this study primarily targeted the utilization of previously inaccessible pores rather than the creation of new unsaturated metal sites, and thus, we found it fitting for linker vacancies to be expressed in this way. Only in a few cases when no methane uptake enhancement was initially observed, the terminal water molecules were removed as an extra measure since they can be deemed to be extraneous to the overall coordination state of the metal clusters

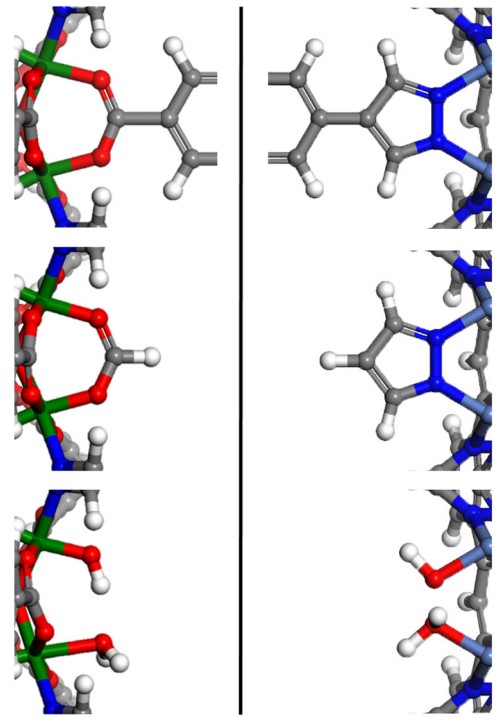

**Fig. 4** Defect introduction schemes for linkers with carboxylate or azolate end groups. Sections of two different MOFs with carboxylate (left) and pyrazolate (right) end groups are shown as examples. The top section shows the original coordination environment for each binding group. The middle section shows replacement with modulators, where we chose formate in place of the carboxylate end group and lone pyrazole anion in place of the pyrazolate end group. The bottom section shows the coordination of water and hydroxidesPlease confirm if all edits in Fig. 4 are correct.Edits in Fig. 4 are correct.

(Supplementary Discussion 2). For MOFs where several different linkers were found (AXUBOL) or distinct coordination environments were observed (KOCWEF)[53] as shown in Supplementary Fig. 1 and Supplementary Fig. 2, each linker or coordination environment was treated separately.

The resulting MOF structures with defects were further optimized using MOPAC. Then, the relaxed configurations were visually inspected and the changes in their unit cell volume with relaxation were tracked. We presumed that the resulting volume difference of $< 5\%$ is sufficient for assuring the feasibility of introducing linker vacancies in the current stage of our research. The results, presented in Supplementary Table 3, show that all of the MOFs show no significant framework collapse with linker vacancy defects.

It is important to understand that the resulting distribution of linker vacancies as expressed by the above methodology is considered to be "correlated". With the application of the periodic boundary condition, the newly created defective unit cell will be replicated infinitely in all three dimensions. Then, the exact same defect configuration will be used throughout the crystal, as shown in Supplementary Fig. 3. A more realistic scenario may be a purely random distribution of defects, where the number of defects per unit cell can vary, resulting in some unit cells with no defects and some with multiple defects. However, such random distribution of defects cannot be directly considered in GCMC simulations as it would at least require an immense unit cell that sufficiently considers all different defect scenarios and mitigates the effects of correlation. Thus, this study primarily reports the case of correlated distribution of defects, which can be directly tested with the current scheme of GCMC simulations. The case of

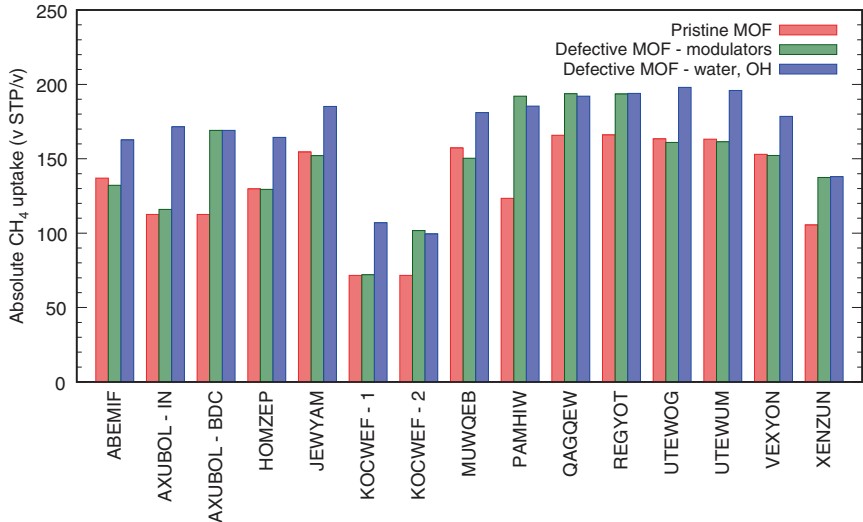

**Fig. 5** Absolute methane uptake amounts of candidate MOFs under different defect scenarios. Absolute methane uptake at 65 bar, 298 K is presented for each candidate MOF. The MOFs are presented by their CSD reference codes. AXUBOL has both IN and BDC linkers incorporated into the framework, and KOCWEF has two different coordination states for the same linker

| Table 1 List of MOFs and their defect scenarios leading to notable methane uptake enhancements | | | | | | |
|---|---|---|---|---|---|---|
| **MOF CSD Refcode** | **Modulators used** | **% of linker vacancies** | **Uptake—defect 65 bar (v STP/v)** | **Uptake—pristine 65 bar (v STP/v)** | **Uptake enhancement 65 bar (v STP/v)** | **% of uptake enhancement** |
| PAMHIW[54] | Formate | 8.33 | 192.09 | 123.38 | 68.71 | 55.69 |
| PAMHIW | Water, OH | 8.33 | 185.43 | 123.38 | 62.05 | 50.29 |
| AXUBOL (IN) | OH (water X) | 5.56 | 171.52 | 112.56 | 58.95 | 52.37 |
| AXUBOL (BDC) | Formate | 5.56 | 169.13 | 112.56 | 56.56 | 50.25 |
| AXUBOL (BDC) | Water, OH | 5.56 | 169.04 | 112.56 | 56.48 | 50.17 |
| KOCWEF (1) | OH (water X) | 6.25 | 107.03 | 71.65 | 35.38 | 49.38 |
| HOMZEP[55] | Water, OH | 8.33 | 164.35 | 129.78 | 34.57 | 26.64 |
| UTEWOG[56] | Water, OH | 6.25 | 197.94 | 163.43 | 34.50 | 21.11 |
| UTEWUM[56] | Water, OH | 6.25 | 195.87 | 163.19 | 32.68 | 20.03 |
| XENZUN[57] | OH | 6.25 | 138.01 | 105.43 | 32.57 | 30.90 |
| XENZUN | Formate | 6.25 | 137.38 | 105.43 | 31.95 | 30.30 |
| JEWYAM[58] | Water, OH | 6.25 | 185.23 | 154.46 | 30.77 | 19.92 |
| KOCWEF (2) | Formate, tetrazolate | 6.25 | 101.78 | 71.65 | 30.13 | 42.05 |
| REGYOT[59] | Water, OH | 6.25 | 193.98 | 166.00 | 27.98 | 16.85 |
| QAGQEW[60] | Formate | 6.25 | 193.73 | 165.82 | 27.91 | 16.83 |
| KOCWEF (2) | Water, OH | 6.25 | 99.38 | 71.65 | 27.73 | 38.69 |
| REGYOT | Formate | 6.25 | 193.50 | 166.00 | 27.49 | 16.56 |
| QAGQEW | Water, OH | 6.25 | 192.09 | 165.82 | 26.27 | 15.84 |
| ABEMIF[61] | Water, OH | 6.25 | 162.62 | 136.95 | 25.67 | 18.74 |
| VEXYON[62] | Water, OH | 6.25 | 178.52 | 152.98 | 25.54 | 16.70 |
| MUWQEB[63] | Water, OH | 6.25 | 181.02 | 157.20 | 23.81 | 15.15 |
| Entries are presented in the order of highest uptake enhancement to lowest uptake enhancement in (v STP/v) at 65 bar, 298 K | | | | | | |

purely random distribution of defects is still considered via indirect methods and is presented in Supplementary Discussion 3.

**Methane uptake enhancement of final candidate MOFs.** To study the effects of linker vacancies on the adsorption properties, $CH_4$ GCMC simulations were then conducted at $T = 298$ K on the relaxed pristine and defect configurations of the 13 candidate MOFs. Blocking feature of the graphics processing unit (GPU) code was still used to detect for the possible existence of inaccessible pores even after the introduction of linker vacancies. The

resulting changes in absolute uptake of methane at $P = 65$ bar are presented in Fig. 5. For all 13 candidate MOFs, at least one of tested defect scenarios resulted in significant enhancement in methane uptake, implying that previously inaccessible pores within the framework have become available for methane adsorption. Also, as can be seen from Supplementary Discussion 3, while there is small overall degradation in the enhancement across the MOFs, this enhancement trend still holds even for the random distribution of defects.

Table 1 lists the different defect scenarios of 13 final candidate MOFs that has led to methane uptake enhancements. Majority of the MOFs required the coordination of smaller coordination

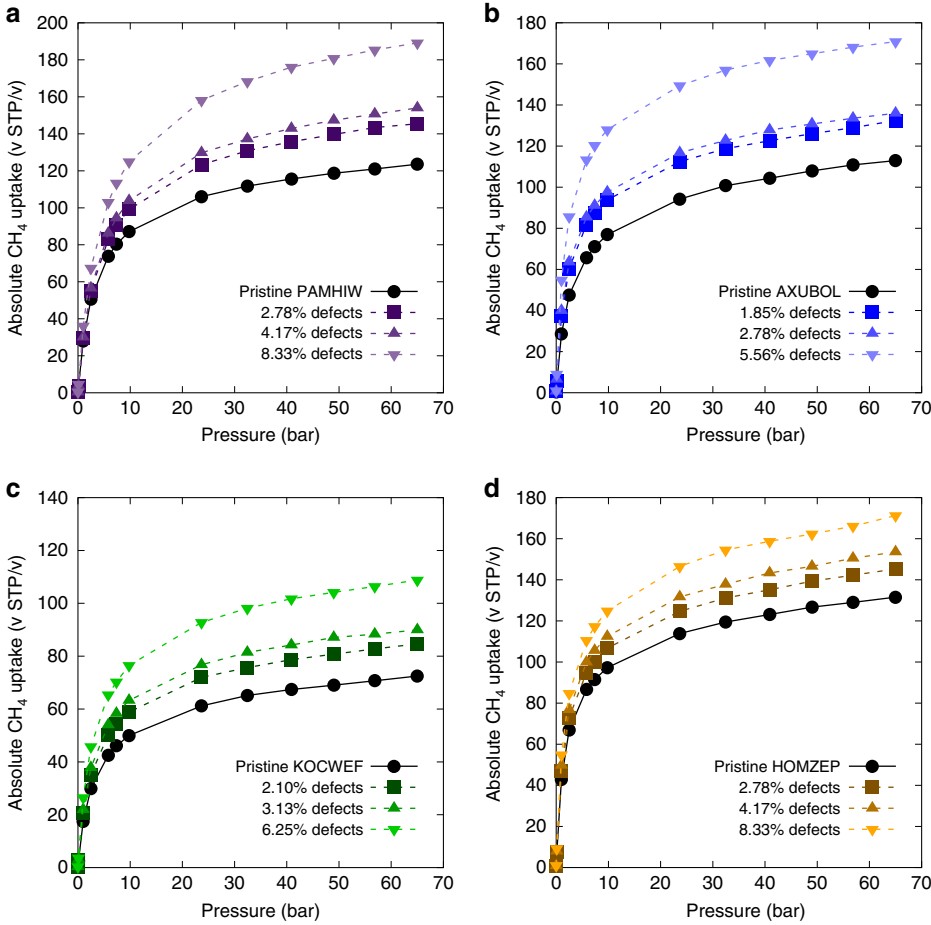

**Fig. 6** Methane adsorption isotherms of four candidate MOFs with the highest uptake enhancement. Methane adsorption isotherms at $T = 298$ K are presented for candidate MOFs with the highest enhancement observed at 65 bar. **a** shows PAMHIW, **b** shows AXUBOL, **c** shows KOCWEF, and **d** shows HOMZEP. Defect sites were coordinated with water and hydroxyl modulators

groups (i.e., water and OH) to experience an enhancement in their methane uptake. These MOFs were found to be comprised of small tridentate linkers or exhibiting a complex topology, where replacing a single linker with bulky modulators was insufficient in increasing the aperture into the inaccessible pores for methane diffusion. For MOFs that experienced enhancement with the modulators, linkers were bidentate, requiring fewer modulators and thus less obstruction, or the original linker was large enough so that the newly created apertures are sufficiently wide enough for methane diffusion even with the modulators.

Although MOFs with absolute enhancement values of higher than 50 (v STP/v) at 65 bar were chosen as candidates in our screening process, one can easily note that the final enhancements were lower than 50 (v STP/v) in majority of the defect scenarios, with the lowest being 23.81 (v STP/v) for MUWQEB. We attribute this to two factors: (1) removal and rearrangement of framework atoms during energy minimization and (2) presence of multiple inaccessible pores within the unit cell. During defect creation and energy minimization, one linker is removed and the rest of the framework atoms are adjusted to achieve the lowest energy configuration in total. Removal of the linker reduces a portion of preexisting interaction sites, and further rearrangement of the atoms can cause reduction in the predicted volume of inaccessible pores. Also, multiple inaccessible pores can be compartmentalized away from one another within a single unit cell of a given MOF. It then requires additional vacancy defects to be present in the vicinities of unaffected inaccessible pores for the resulting uptake enhancement to better match the initially

projected values from the screening process. This presents the balance between striving for the maximal attainable enhancement and stabilizing a large number of defects within the framework. We chose to maintain a conservative stance on this matter by restricting the proportion of linker vacancies. It is interesting to point out that out of the final 13 candidate MOFs, 9 were found to have sodalite topologies with tridentate linkers. For these MOFs, beta cages of their sodalite structure were found to be inaccessible toward methane, and only when the linker vacancies are created these inaccessible pores can become accessible.

To further showcase the effects of linker vacancies on the candidate MOFs, full adsorption isotherms under several different defect proportions for the four candidate MOFs with highest enhancement at $P = 65$ bar are presented in Fig. 6. In controlling the defect proportions, the unit cells were further expanded by a factor of 2 and 3 prior to the introduction of a single linker vacancy. Because the total number of linkers per unit cell can be different for each candidate framework, different defect proportions are used for each MOF. Nonetheless, persisting trend of enhancement in their methane adsorption isotherms (throughout all pressure regimes) was observed even for the lowest proportion of defects tested, suggesting that an even lower proportion of defects can lead to observable changes in the adsorption isotherms of our candidate MOFs.

**Further confirmations of inaccessibility phenomenon.** To provide evidence that the enhancement phenomenon is unique to the candidate MOFs obtained via our screening methodology, five

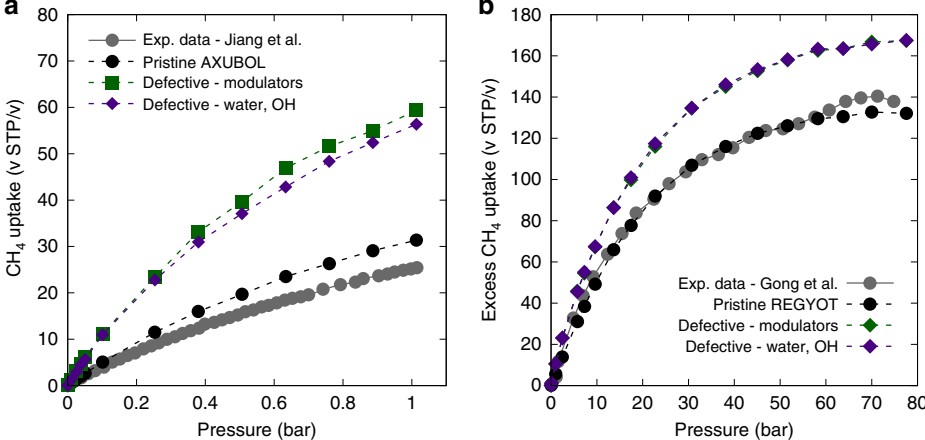

**Fig. 7** Comparison of GCMC simulation results with available experimental data for AXUBOL and REGYOT. **a** Methane adsorption isotherms of AXUBOL at 273 K in the low-pressure regime. **b** Methane adsorption isotherms of REGYOT at 298 K in the high-pressure regime. Gray isotherm is the experimental methane adsorption data from previous references for the respective candidate MOFs. GCMC simulation results of the pristine case in our simulations are found to closely match the available experimental data

MOFs from the CoRE MOF database that did not satisfy our screening criteria were randomly selected and checked for changes in their methane uptake profiles with linker vacancy defects expressed with water and hydroxides. Additionally, we conducted the same tests on UiO-66 to show that the observed enhancement for defective UiO-66 in a previous study was highly dependent on the mesopore formation. Adsorption isotherm data at 65 bar and 298 K (Supplementary Fig. 4) show that all six of the tested MOFs do not show a clear trend of enhancement, thus demonstrating the utility of our screening criteria. We also tested the sensitivity of our findings on the force field utilized by repeating the methane adsorption calculations under the same conditions with DREIDING force field applied on the framework. Results are presented in Supplementary Fig. 5, which should be closely compared with the original results in Fig. 6 produced using the universal force field (UFF). There are slight differences in the methane uptake values for each MOF in using different force fields. Nevertheless, the same enhancement trend with linker vacancies persists for all MOFs, verifying that enhancement phenomenon is robust to changes in the force field parameters used in the simulation.

In addition, to test for the existence of inaccessible pores using a different gas molecule, GCMC simulations were conducted with $H_2$ (instead of $CH_4$), and the results at 65 bar and 77 K are presented in Supplementary Fig. 6. Interestingly, one of the candidate MOFs showed a different enhancement pattern compared to the case of methane. In the case of AXUBOL, the $H_2$ uptake amount for the pristine and defect configurations was nearly the same, indicating that the inaccessibility phenomenon shown in $CH_4$ does not extend to $H_2$. This shows that inaccessibility phenomenon can be highly sensitive to the gas molecule of interest. Each candidate MOF would then have a characteristic effective pore aperture size at which the secluded pores become inaccessible toward the gases of bigger sizes.

To further confirm the existence of inaccessibility and potential for enhancement with linker vacancies, comparisons were made between previously published experimental methane adsorption data with our simulated isotherms. Of the final 13 candidate MOFs, AXUBOL and REGYOT were found to have experimental methane isotherms available in their original publications and as such, the isotherm data for these two materials were compared as shown in Fig. 7. In both cases, the experimental adsorption isotherm data are found closer to the simulation isotherms of pristine MOFs with the inaccessible pores blocked appropriately

in the simulations. This serves as a strong indication that methane-inaccessible pores do indeed exist for the tested MOFs, and thus it is possible to induce an uptake enhancement by introducing linker vacancies. Also, for better understanding of the inaccessibility phenomenon and enhancement of methane uptake thereafter, methane energy contours of AXUBOL and REGYOT with and without linker vacancy defects were produced and are presented in Fig. 8. In both MOFs, a single linker within the unit cell have been replaced with water and hydroxides to open up the inaccessible pore detected in our simulations. Resulting defect structure for each MOF shows notable expansion of the energy-favorable volume within the unit cell, where the inaccessible pores have merged with the preexisting adsorption channels.

## Discussion

It is important to rigorously consider the viability of experimentally reproducing the linker vacancy scenarios of candidate MOFs that we have proposed in this work. Previous researches have shown that creating vacancy defects with minimal modulation schemes may require immense free energies[28, 40], and linker vacancy defects can greatly compromise the mechanical stability of resulting MOFs[64]. Considering the nature of our current research, we deemed it is not yet necessary for us to perform thorough calculations of chemical feasibility and mechanical stability of the suggested defective MOF structures with linker vacancies. However, simply maintaining the fraction of linker vacancies to be lower than what has been reported for UiO-66 is most likely insufficient in guaranteeing the possibility of experimentally creating linker vacancies. Thus, it is of crucial importance to address these issues thoroughly in future researches to follow.

With close to 70,000 MOFs reported to date[2], it is virtually impossible to synthesize and judiciously introduce defects to every framework, then test for changes in the adsorption properties. In this work, we have demonstrated that rational defect engineering using computational simulations can identify materials that can lead to significant enhancement in methane adsorption properties with small proportions of linker defects. As such, we have presented a blueprint in which the experimentalists can review previously synthesized MOFs and re-synthesize them with linker defects to obtain materials that have significantly different properties from the parent materials. The use of rational defect engineering presented in this study introduces a new dimension to the current scheme of MOF studies, where precise

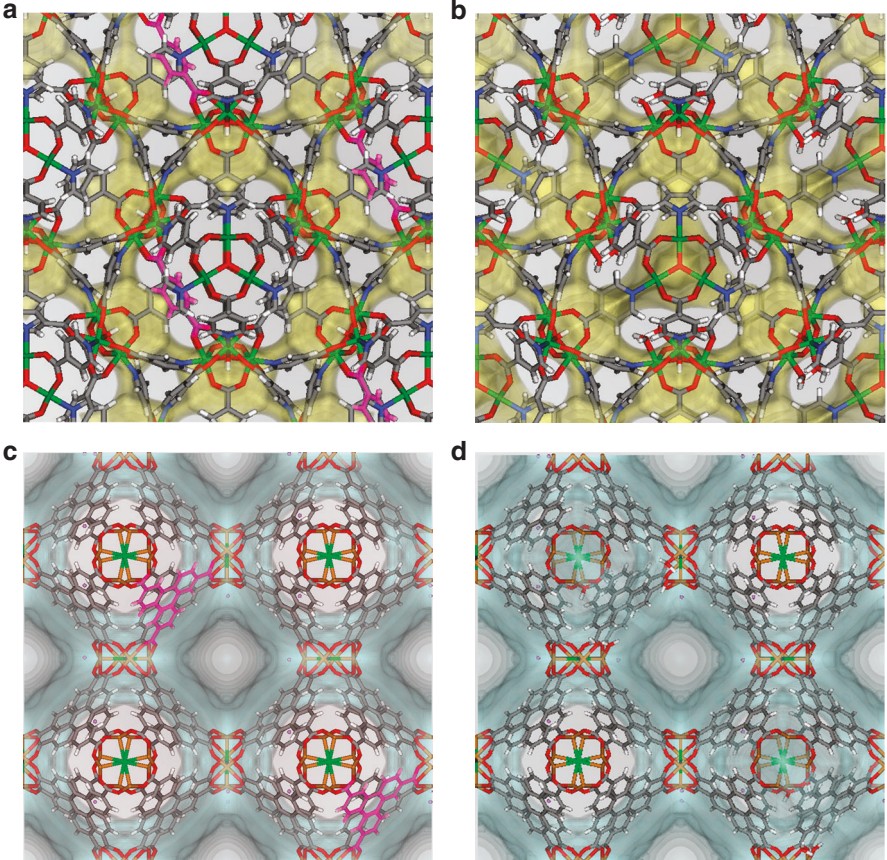

**Fig. 8** Methane energy contours of AXUBOL and REGYOT before and after defect introduction. The yellow contours for **a** pristine AXUBOL and **b** defective AXUBOL highlight the methane energy surface with 0 kJ mol$^{-1}$. The blue contours for **c** pristine REGYOT and **d** defective REGYOT highlight the methane energy surface with 0 kJ mol$^{-1}$. The energy contours were produced for $T = 298$ K. In **a** and **c**, linkers highlighted in magenta correspond to the removal of a single linker from one unit cell. (white: hydrogen, gray: carbon, blue: nitrogen, red: oxygen, dark green: nickel, light green: chlorine, orange: copper, purple: lithium)

control over distribution and concentration of defects can lead to desirable changes and perhaps new application areas for existing MOFs.

Although the focus of this work was on methane for its simplicity, the core concept can be readily extended to other gas molecules, suggesting the vast possibilities of different schemes in which linker vacancy defects can induce the uptake enhancement for a given MOF-adsorbate pair. In saying that, we also note the possibility of even more significant performance enhancements for gas molecules with dipole moments such as $CO_2$ and $H_2O$. We predict their heightened interactions with newly created unsaturated metal sites, coupled with merging of new pores for adsorption, can lead to even higher degree of enhancement in their adsorption profiles. Moreover, the excavated pores can be used for selective adsorption of certain gas components within a gas mixture that can potentially enhance the selectivity of smaller molecules over a larger one via careful design.

Finally, it is important to recognize that excavation of inaccessible pores via defect engineering is only one out of many different ways in which the previously unexploited pores can be utilized for enhanced performance. We envision using the idea of inaccessibility to have potential for other applications (e.g., long-term trapping of gas molecules for flexible MOFs with significant number of inaccessible pores) that can lead to exciting new developments for some of these synthesized MOFs. Accurate prediction of the presence of inaccessible pores and development of novel methods for their utilization introduces a new dimension to the current scheme of MOF research, where the previously

synthesized MOFs can undergo further chemical modifications to reach their maximum potentials in various applications.

## Methods

**Grand canonical Monte Carlo simulations**. All GCMC simulations and Henry coefficient calculations were conducted using a high-throughput GPU code developed by Kim et al.[65, 66] UFF[67] was chosen as the force field parameter for screening and adsorption simulations of the MOFs. DREIDING force field[68] was also adopted to check for reproducibility of methane adsorption data of the final candidates. Methane was described using the TraPPE force field[69], and hydrogen interaction parameters were taken from Buch et al.[70] Lorentz–Berthelot mixing rule was applied in describing the interaction between different atoms. In the case of hydrogen gas simulations at $T = 77$ K, Feynman–Hibbs correction was applied. All framework atoms were assumed to be rigid in our simulations. A Lennard–Jones cutoff radius of 12.8 Å was imposed in the calculations, and an energy grid with 0.15 Å spacing in all dimensions was constructed for each MOF. Henry coefficient calculations were conducted with the Widom particle insertion method, and pure component GCMC simulations of methane and hydrogen were performed with 50,000 Monte Carlo cycles.

**Inaccessible pore blocking algorithm**. To identify inaccessible regions within the MOFs, a flood fill algorithm was utilized, hereby referred to as the "blocking feature", shown in Supplementary Fig. 7. Using the energy grid generated from the GPU code, the blocking feature performs a flood fill on low-energy regions of the framework, where adsorbates are likely to be found. In identifying the low-energy regions for adsorption, a previously used energy threshold of 15 $k_B T$ (with $T = 298$ K) was utilized[66], where grid points with higher energy was considered inaccessible (marked "1") toward the adsorbate, and the rest were considered accessible (marked "0"). This specific threshold value was further rationalized after testing several other energy thresholds (i.e., $k_B T = 6, 9, 12, 18, 21, 24$) and noting no changes in the methane uptake amount of our final candidates, as shown in Supplementary Table 4. From the flood fill results, only the main adsorption channels that are endlessly connected over the periodic boundaries were considered

accessible, and the secluded pockets of energy low regions were considered inaccessible (0 → 1). When the blocking feature is enabled, grid points determined to be inaccessible are re-assigned fictitiously large energy values.

**Energy minimization of MOFs using semi-empirical methods**. Considering the large number of frameworks and defect scenarios studied, this study adopted a semi-empirical method rather than an ab initio method to efficiently consider all structures within a reasonable period of time. All energy minimization calculations of the MOFs were performed using MOPAC 2016, with PM7 given as the semi-empirical Hamiltonian and GNORM = 10. Validation of MOPAC and PM7 is detailed in Supplementary Discussion 4 with comparisons against DFT energy minimization results for 50 select MOFs.

**Data availability**. Chemical formulas and unit cell diagrams of the candidate structures are presented in Supplementary Note 1. Crystallographic information files of the candidate structures in their pristine and defective form after semi-empirical MOPAC relaxations are also provided as Supplementary Datasets. All other data that support the findings of this study are available from the authors upon reasonable request.

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

## Acknowledgements

S.C. and J.K. were supported by Basic Science Research Program through the National Research Foundation of Korea funded by the Ministry of Science, ICT, & Future Planning under grant no. 2017R1A2B4004029, and by the BK21 Plus Program funded by the Ministry of Education (MOE, Korea). G.T. thanks the Lepoldina—Nationale Akademie der Wissenschaften for a postdoctoral scholarship. We greatly thank Prof. Jeffrey Long (UC Berkeley) for the useful discussions. We also thank Dr. James Stewart for providing us with an academic license to MOPAC 2016.

## Author contributions

S.C. performed computational simulations and prepared the SI. G.T. suggested the defect coordination schemes and helped with the analysis. J.K. formulated the project. All authors contributed to exchanging ideas and writing the manuscript.

## Additional information

**Competing interests:** The authors declare no competing financial interests.

