## [Peer Review File · Nature Communications]

Reviewers' comments:

Reviewer #1 (Remarks to the Author):

This work is an interesting new approach, but the data obtained and the methods used do not fully back the conclusions reached. Major points to be addressed before it can be considered for publication:

- The semi-empirical method used is not well validated, neither in this work nor in the published literature. The authors should first validate this methodology before using it. Our own experience shows that semi empirical methods (and PM7 in particular) is quite poor for describing MOFs.
- It is not clear how the site for introduction of defects was selected. The authors should clearly state what their assumption is: random defects or correlated order? What about defects being introduced that do not affect the blocked sites/pockets? Surely that must happen. Do the authors average (as they should) over all possible positions of the defect?

Minor comments:

- Some of the literature on defects in MOFs and their impact on properties is not well cited: for example, papers like (DOIs): 10.1002/anie.200806063, 10.1021/ja404514r, 10.1038/nchem.2691
- "we combine two previously unrelated concepts of "linker vacancy" and "inaccessibility" (page 5): this is not new, and several authors have established this link before, and many studies on defects and adsorption in MOFs have done exactly that. Remove this claim.
- On figures 3 and 4, the large number of points near $x=y$ (and the fact that they are plotted as very large symbols) means the reader does not gain a good understanding of the relative density of points near the $x=y$ line with respect to the rest of the graph. Possibly a heat map plot would give a better indication.
- Defect rates can be much higher than 1/12, as shown by the Telfer group (DOI: 10.1021/acs.chemmater.5b04306)... this should be discussed, rather than quote 1/12 as a "universal" limit for MOFs.
- For the sake of reproducibility, the authors should include representative input files as supporting information (or make them available online).

Reviewer #2 (Remarks to the Author):

The paper by Kim and coworkers utilized a computational method to explore the possibility of improving the gas adsorption capacity of MOFs by defect engineering. The author reasoned that linker vacancy defects could possibly expose the otherwise inaccessible pores of MOFs, which could in turn improve the methane storage capabilities. Therefore, 13 candidates were selected out of 11,558 structures from MOF databases by high-throughput computational screening. Defected models were built for candidate MOFs by partially replacing linkers with modulators or solvents. Grand canonical Monte Carlo simulations further demonstrated the contribution of defects on the methane storage capacities. Overall, this paper provides a quite unique perspective to explore the effect of defects on the MOF performance. It will guide future researches on the rational defect engineering to maximizing

the potential of existing MOFs. Therefore, I suggest publishing this paper after addressing the following questions.

1. The CoRE MOF database contains 5109 structures and the DFT-minimized CoRE MOFs have 1340 structures (Chem. Mater. 2014, 26, 6185–6192; Chem. Mater. 2017, 29, 2521–2528). The structures presented in DFT-minimized CoRE MOFs could also be presented in CoRE MOF database. The total structures with unique CCDC reference code is definitely less than 11558 (page10 line 188). Are there any other databases that were used in this work but not cited?
2. The position of the removed linker should be clarified. In addition, when multiple linkers were removed, their relative position should also be indicated. If a missing linker defects present in the vicinities of inaccessible pore, the pore will be labeled accessible by the flood fill algorithm. The enhanced adsorption in Figure7 is strongly depending on the position of the linker vacant defects. Therefore, linker removal strategy should be explained in details.
3. It would be more relevant to simulate the methane uptake of the defected MOFs by removing the terminal waters, as the terminal water on most 2+ metals could be removed during the activation of MOFs.
4. ABEMIF, UTEWOG, VEXYON, and XENZUN are anionic frameworks with metal cations/ $[(\text{CH}_3)\text{NH}_2]^+$ in the cavity. The cations could affect the gas adsorption, however, they are omitted in the CoRE MOF database. Are they considered in the simulation?

Reviewer #3 (Remarks to the Author):

In this paper, Chong et al. report a new strategy for improving existing MOFs for gas sorption: finding frameworks which contain inaccessible void volumes that could be connected to the accessible volume by introducing ligand vacancies. They then carry out a screening on tens of thousands of reported MOF crystal structures and find that significant improvement in CH₄ uptake could be achieved in 13 of these structures. The authors report an ingenious strategy, and follow it through to its logical theoretical conclusion, yielding suggestions that could be directly experimentally tested.

I have a few technical questions:

- the authors refer to an energy of kT throughout: what T are they using?
- the authors relax the structures using MOPAC to check for structural stability post defect inclusion, and find in one case that the structure changes by about 15%V. Did the authors also relax the non-defective structure in MOPAC to check that the changes are not due to the different simulation protocol?

The text is in general clear and explains the ideas well. Unfortunately, the authors' extremely diverse vocabulary and elaborate style sometimes impeded my ability to understand their intended meaning: for example, is a 'volume offset' the percentage change in volume or in lines 101-103, do the authors believe that only UiO-type MOFs could be affected by defects, or do they think that other people could come to this conclusion? If the authors were to go through and refocus their text to simplify some of the writing I think it would greatly help readers.

Two other aspects of the manuscript are not ideal for readers. A few times I encountered abbreviations and symbols which were not yet defined (BDC, IN, XSHV, KH): in some cases I had to consult the SI, and in one or two, I am still unclear as to their meaning (XSHV?). Additionally, although I greatly appreciated the inclusion of diagrams of the crystal structures in the SI, the perspectives sometimes chosen are sometimes not very informative (e.g. AXUBOL on page 10). I

would also very much like if the authors presented their final structures: could they show diagrams of the structures after they have substituted in the defects, and also include CIFs or similar of the structures? At present, figure 5 and 9 are the only figures to give some structural information about the resultant structures.

In this point-by-point response to the Reviewers' comments:

1. Black color - comments from the Reviewers
 2. Blue color - response to the comments
 3. Green color - added in the manuscript with page number provided
-

Reviewer #1 (Remarks to the Author):

This work is an interesting new approach, but the data obtained and the methods used do not fully back the conclusions reached. Major points to be addressed before it can be considered for publication:

- The semi-empirical method used is not well validated, neither in this work nor in the published literature. The authors should first validate this methodology before using it. Our own experience shows that semi empirical methods (and PM7 in particular) is quite poor for describing MOFs.

The reviewer brings up a very important point. Though using a more accurate energy minimization method involving DFT would be ideal, it becomes computationally expensive to perform DFT calculations for all the MOFs and each of their defect configurations. Thus, we have used the semi empirical PM7 method to obtain the energy minimized configurations of the pristine/defective MOFs. There have been several previous studies showing reasonable applicability of PM7 and other semi empirical methods in relaxing MOFs (DOI: 10.1021/acs.jpcc.5b05599, DOI: 10.1021/jp401920y). However, we still agree with the reviewer that semi empirical methods can be quite poor at times compared to other methods involving a higher level of theory, in describing MOFs and measuring their properties. However, through our own validation process presented below, we propose that the usage of PM7 can still be considered suitable within the context of this work.

Instead of performing the DFT calculations ourselves, we utilized a set of 838 DFT-relaxed MOFs published by Nazarian and Sholl et al. (DOI: 10.1021/acs.chemmater.6b04226) to validate the PM7 relaxation results. These are subset of the same structures (the only difference being that they were relaxed using DFT) used within the CoRE MOF database. Of our 13 candidate materials selected through the screening criteria, two MOFs, UTEWOG and UTEWUM, are from the DFT-relaxed MOF database from Nazarian and Sholl. We computed the blocking on/off methane adsorption isotherms at T = 298 K for both of these MOFs with their DFT and PM7 relaxation configurations. As can be seen from the figure below, we see that there is almost perfect agreement in the adsorption isotherms, and the flood-fill algorithm can detect the inaccessible regions consistently under both relaxation schemes.

Since having only two MOFs is too small of sample, we expanded our validation process to include 50 additional MOFs from the DFT-relaxed set. Among the 50 selected MOFs, 6 were chosen to be MOFs with $(K_{H, \text{blocking off}})/(K_{H, \text{blocking on}}) > 1.5$ that we have identified from our screening process, meaning that they are bound to possess inaccessible pores. Rest of the structures (44) were chosen at random. In the random selection of these MOFs, only the structures with largest free diameter of higher than 3.5 Å under the use of Zeo++ (DOI: 10.1016/j.micromeso.2011.08.020). This was also done in part to guarantee porosity towards methane, and also for the fair evaluation of blocking phenomenon under different relaxation schemes. Each of the selected MOFs was relaxed from their unrelaxed experimental configurations using PM7 with MOPAC under the same conditions specified in our manuscript.

The PM7-relaxed configurations of 50 selected MOFs were visualized to search for any unwanted structural collapse or bond detachment, same as what we would have done during our actual screening process. The visual inspections revealed that two of the chosen MOFs showed bond detachment around the metal cluster (GIFKIP, RABHAZ). These invalid structures were consequently removed from further considerations in our validation process, as they would also have been removed during our screening process had they been considered as candidates.

On the remaining MOFs and their PM7-relaxed and DFT-relaxed configurations, methane GCMC simulations at 298 K was conducted twice, once with the blocking feature enabled and once disabled. The results for the two relaxation schemes was then compared at $P = 65$ bar, and the scatter plot is presented below:

There unfortunately still exist cases where large discrepancies exist between the two methods, for structures such as WALBOC ($\Delta = 39.578 \text{ cm}^3/\text{cm}^3$) and YEGCUJ ($\Delta = 31.820 \text{ cm}^3/\text{cm}^3$). Most other structures show very good agreement, however, allowing us to observe an R^2 value of 0.963 when data points are fitted to $y = x$.

Finally, we also compared the list of MOFs containing inaccessible pores as detected by our flood-fill algorithm after structure relaxation:

MOFs with Inaccessible Pores, Sholl DFT	Uptake difference with blocking at P = 65 bar (cm^3/cm^3)	MOFs with Inaccessible Pores, PM7	Uptake difference with blocking at P = 65 bar (cm^3/cm^3)
BAEDTA	20.877	BAEDTA	21.847
ECAHAT	23.173	ECAHAT	24.115
IPICUG	35.441	IPICUG	35.982
QEFNAQ	36.949	QEFNAQ	38.017
TETZID	26.489	TETZID	24.015

XADGAM	29.461	XADGAM	28.800
ZIDDIB	27.707	ZIDDIB	23.884
		ATOXEN	7.376
		LUVTEC	0.983

Note that first seven structures listed for both relaxation schemes are identical. What may seem concerning is the additional inclusion of two MOFs by PM7, ATOXEN and LUVTEC, that were never identified from the DFT relaxation scheme. However, the uptake difference with and without blocking at 65 bar for these MOFs is $7.376 \text{ cm}^3/\text{cm}^3$ for ATOXEN, and $0.983 \text{ cm}^3/\text{cm}^3$ for LUVTEC. These are negligible differences that would cause these structures to be filtered out during the course of our screening process. All in all, PM7 can correctly predict all seven MOFs with significant volume of inaccessible pores found from the DFT relaxation method, and our additional screening criteria would limit us from falsely including any other structures that have developed residual amounts of inaccessible regions from PM7 relaxation.

Through our validation process, we suggest that PM7 can be an acceptable method of energy minimization within the context of our research. PM7 relaxation shows plausible agreement with DFT relaxation in methane GCMC simulations for 50 selected MOFs akin in their porosity towards methane with our candidates. Issues that may arise from the inaccuracies of PM7, such as incorrect bonding expression or formation of residual amounts of inaccessible volume, cannot hinder our screening results as the screening criteria used in our study limit MOFs susceptible to such issues from making the final candidate list.

We have made following changes and additions in the manuscript and supplementary materials with regards to the validation of using semi empirical PM7 method.

Page 8: "Validation of MOPAC and PM7 is detailed in the Supplementary Information with comparisons against DFT energy minimization results for 50 select MOFs."

SI: Section 2 (SI pg. 2-5)

- It is not clear how the site for introduction of defects was selected. The authors should clearly state what their assumption is: random defects or correlated order? What about defects being introduced that do not affect the blocked sites/pockets? Surely that must happen. Do the authors average (as they should) over all possible positions of the defect?

Again, this is an important point raised by the reviewer.

In the manuscript, a single linker from the unit cell is being removed to observe any uptake enhancement, and this linker was chosen purely at random. For the candidate MOFs, high symmetry formed by the linkers and metal nodes leads to the same uptake enhancement being observed for all possible sites of linker vacancy within the unit cell. We have tested this for all of the 13 final candidates, where four different defect configurations that only differ in the relative location of the missing linker defect were created and tested for any differences in methane uptake enhancement. Results, presented below for each candidate MOF containing water & OH coordinating defects, indicate that there exists virtually no error or deviation in uptake between different positions of the linker defect within the unit cell.

The defect introduction scheme used in our manuscript can be considered as being “correlated”, in the sense that the linker vacancy was introduced to the unit cell, and this image was replicated infinitely in all 3 dimensions to express the defect crystal as a whole. This results in every single unit cell within the material experiencing the same uptake enhancement in a very ordered and correlated manner. However, this can be considered highly unrealistic, and the highly correlated defect distribution guarantees maximum uptake enhancement will be reached under the given defect proportion.

During the revision process, we newly considered the extreme case of purely randomized distribution of defects, meaning that the number of defects per unit cell would no longer be fixed and can hold any value between 0 up to total

number of linkers per unit cell (under the constraint of fixed defect percentage in the entire crystal). This means that some unit cells can be defect-free without any uptake enhancement, whereas other unit cells can contain one or more defects leading to uptake enhancement. We believe this may be a more realistic way of considering the defect distribution into account. However, such randomness in material structure cannot be effectively taken into account in the GCMC simulations due to inevitably large supercell size that is required in including all representative proportions of defect scenarios within individual unit cells. As such, we assumed that the overall adsorption properties can be divided into appropriate linear combination of different unit cells (with different defect proportions) and calculated the uptake enhancement by simply considering the relative proportions of pristine and defect unit cells. The weights given to each of the defective unit cells (e.g. percentage of unit cells having 0, 1, 2, 3, ... defects) were based on how likely these were to be generated based on random removal of defects.

The two defect distribution schemes are shown above. A comparison of the two defect distribution schemes is made in the below bar graph, which shows the uptake enhancement of each of the 13 candidate MOFs as predicted by the correlated defects and randomly distributed defects. Proportion of linker vacancy used for calculation is different for each candidate due to the difference in the number of linker per unit cell. These are found in Table 1 of the manuscript.

The results show small differences between the two defect expression schemes for most of the candidate MOFs. This can be explained as follows: In our analysis, we have kept the defect proportion relatively small (~ 2 to 8% of the total number of linkers). And as such, even in the “correlated” defective scheme, there still are many inaccessible regions that have not opened up due to the small proportion of linker defects. Consequently, the difference between correlated and random defective distribution, considering that not all the inaccessible regions were opened up in the first place, is not too big such that the overall message of the paper will be affected.

We would like to add that one key reason on why we kept the linker defect percentage to be relatively small in the first place was to not “oversell” our results. That is, surely with even higher defect percentages (e.g. 25%), we could have reported larger enhancement values. However, then there is an issue of framework stability that becomes more difficult to justify in the screening work, which might invite a whole level of skepticism towards the entire work. Moreover, it is conceivable that some of the readers (when they see a very high defective percentage value such as 25%) might think that it is not the inaccessible regions but the void space left by the linkers that is playing more of a crucial role in the enhancement. And we wanted to clearly deliver the message that this was not the case and that our mechanisms for enhancement was different.

An explanation for the two defect expression schemes is given in the main manuscript, and results for random defect distribution case is presented in Supporting Information. In revising the manuscript/SI, we have decided to include the correlated defect results in the manuscript and the randomized result in the SI for the following reason: because the correlated defect results were based on actual GCMC simulation data (and not linear combination of GCMC simulations)

and require less approximations, this was deemed to be more appropriate in the main manuscript. Moreover, the difference in the enhancement values between the two schemes is not too different and therefore the overall message of the results remains the same. However, we think it is very important to present both the correlated and the randomized results as the readers might be interested in the differences between the two, and as such, we made sure to include this data in the Supporting Information.

With regards to the next point, as the reviewer points out, one could suspect that some defects being introduced may not affect the blocked pockets. For the final candidates presented in the manuscript, however, removal of any linker in the pristine unit cell results in the incorporation of inaccessible pores to the main channel and a significant uptake enhancement. The screening criteria of K_H ratio > 1.5 and blocking uptake difference $> 50 \text{ cm}^3/\text{cm}^3$ was rigorous enough to identify only the MOFs with abundance of inaccessible pores, to the point where all of the linkers are bordering an inaccessible pore. It is only when one attempts to add additional defects into the already defective unit cell that a given defect may not be able to affect any new blocked pockets (due to the inaccessible region already being open). Such cases were not directly handled in our manuscript, as even the highest defect proportion used for simulations (8.333% or 1/12) did not require multiple defects to be present in the same unit cell.

Page 16: “It is important to understand that the resulting distribution of linker vacancies as expressed by the above methodology is considered to be “correlated”. With the application of the periodic boundary condition, the newly created defective unit cell will be replicated infinitely in all three dimensions. Then, the exact same defect configuration will be used throughout the crystal, as shown in Figure S7. A more realistic scenario may be a purely random distribution of defects, where the number of defects per unit cell can vary, resulting in some unit cells with no defects and some with multiple defects. However, such random distribution of defects cannot be directly considered in GCMC simulations as it would at least require an immense unit cell that sufficiently considers all different defect scenarios and mitigates the effects of correlation. Thus, this study primarily reports the case of correlated distribution of defects, which can be directly tested with the current scheme of GCMC simulations. The case of purely random distribution of defects is still considered via indirect methods and is presented in the Supporting Information.”

Page 17: “Also, as can be seen from Figure S8 and Table S7, while there is small overall degradation in the enhancement across all of the MOFs, this enhancement trend still holds even for the random distribution of defects.”

Figure S8:

Table S7:

MOF CSD Refcode	CH ₄ uptake enhancement for each defect rate (cm ³ /cm ³)		
	8.33% defects	12.50% defects	25% defects
ABEMIF	27.457	35.944	49.254
AXUBOL	44.672	51.374	56.163
HOMZEP	31.619	43.440	62.876
JEWYAM	29.613	38.766	53.121
KOCWEF	36.374	46.547	58.488
MUWQEB	22.400	29.324	40.183
PAMHIW	40.208	49.564	60.099
QAGQEW	30.661	40.297	52.204
REGYOT	33.476	44.206	57.571
UTEWOG	34.711	45.440	62.266
UTEWUM	28.119	36.811	50.442
VEXYON	24.345	31.869	43.671
XENZUN	32.472	42.509	58.251

SI: Section 9 (SI pg. 14-17)

Minor comments:

- Some of the literature on defects in MOFs and their impact on properties is not well cited: for example, papers like (DOIs): 10.1002/anie.200806063, 10.1021/ja404514r, 10.1038/nchem.2691

We thank the reviewer for these suggestions and have inserted them as part of References as shown below:

Ref 24: Farrusseng, D., Aguado, S. & Pinel, C. Metal-Organic Frameworks: Opportunities for Catalysis. *Angew. Chemie Int. Ed.* 48, 7502–7513 (2009).

Ref 32: Wu, H. et al. Unusual and Highly Tunable Missing-Linker Defects in Zirconium Metal–Organic Framework UiO-66 and Their Important Effects on Gas Adsorption. *J. Am. Chem. Soc.* 135, 10525–10532 (2013).

Ref 26: Bennett, T. D., Cheetham, A. K., Fuchs, A. H. & Coudert, F.-X. Interplay between defects, disorder and flexibility in metal-organic frameworks. *Nat. Chem.* 9, 11–16 (2016).

- "we combine two previously unrelated concepts of "linker vacancy" and "inaccessibility" (page 5): this is not new, and several authors have established this link before, and many studies on defects and adsorption in MOFs have done exactly that. Remove this claim.

We have changed the sentence to the following: we combine "linker vacancy" and "inaccessibility", and omit the words "two previously unrelated concepts" from the texts.

Page 5: In this study, we combine two concepts of "linker vacancy" and "inaccessibility" to pave the way to excavate the inaccessible pores and improve the gas adsorption capacities of existing MOFs.

- On figures 3 and 4, the large number of points near $x=y$ (and the fact that they are plotted as very large symbols) means the reader does not gain a good understanding of the relative density of points near the $x=y$ line with respect to the rest of the graph. Possibly a heat map plot would give a better indication.

This is a great suggestion. However, instead of using a heat map, we have edited the figures such that the population spread is much more clearly shown. In the case of Figure 3, number of data points have been reduced to only the data relevant to the discussion is being presented.

Page 10: “3,481 structures showed no difference in CH₄ K_H with and without blocking. Distribution of the remaining 243 structures with K_H ratio > 1 is shown in Figure 3, and 118 MOFs were found to exhibit CH₄ K_H ratio of higher than 1.5.”

New Figure 3:

New Figure 4:

- Defect rates can be much higher than 1/12, as shown by the Telfer group (DOI: 10.1021/acs.chemmater.5b04306)... this should be discussed, rather than quote 1/12 as a "universal" limit for MOFs.

We agree with the reviewer that it is possible that some of these MOFs can contain more than 1/12 linker defects. We were not implying that this was a universal limit but we can see how the portion of the texts can be construed in that manner, and as such made changes to show that this "limit" can be indeed more flexible as well as including the reference from the Telfer group.

Disregarding the stability of the MOFs, being able to have more linker defects than 1 out of 12 leads to a further increase in uptake enhancement for most of our candidate MOFs, so we were actually trying to be parsimonious with our claim by using 1/12 as a "limit".

Also, with the implementation of a new evaluation method for randomly distributed defects, methane uptake enhancement under several other defect rates have been calculated and are presented in the SI.

Page 14: "It should be noted, however, that even higher linker defect rates can be experimentally observed in MOFs,⁵⁸ and hence it may also be possible for the candidate MOFs to be engineered to withstand higher defect rates than considered here."

SI: Section 9, Table S7 (SI pg. 17)

- For the sake of reproducibility, the authors should include representative input files as supporting information (or make them available online).

This is again a great suggestion. We have made all the xyz files of the top candidate structures available online. We have also included list of CSD reference codes for the “next-best” MOFs and expanded them to 50. With regards to the simulation, our GPU code has not been released yet as we are still tinkering with it. In the future, we will try to release the code, executable files, tutorial, and input files. We would like to point out that the flood fill algorithm that we use to detect blocking regions is not complicated to code (REF: DOI: 10.1021/ct200787v), so from other software (e.g. RASPA), the energy grids generated can be post-processed to detect blocking regions. Finally, free software like Zeo++ (<http://www.maciejharanczyk.info/Zeo++/examples.html>) has the capacity to detect inaccessible regions and as such, the structure input files (with and without defective linkers) can be used to test for connection of inaccessible regions to the main channels.

Reviewer #2 (Remarks to the Author):

The paper by Kim and coworkers utilized a computational method to explore the possibility of improving the gas adsorption capacity of MOFs by defect engineering. The author reasoned that linker vacancy defects could possibly expose the otherwise inaccessible pores of MOFs, which could in turn improve the methane storage capabilities. Therefore, 13 candidates were selected out of 11,558 structures from MOF databases by high-throughput computational screening. Defected models were built for candidate MOFs by partially replacing linkers with modulators or solvents. Grand canonical Monte Carlo simulations further demonstrated the contribution of defects on the methane storage capacities. Overall, this paper provides a quite unique perspective to explore the effect of defects on the MOF performance. It will guide future researches on the rational defect engineering to maximizing the potential of existing MOFs. Therefore, I suggest publishing this paper after addressing the following questions.

1. The CoRE MOF database contains 5109 structures and the DFT-minimized CoRE MOFs have 1340 structures (Chem. Mater. 2014, 26, 6185–6192; Chem. Mater. 2017, 29, 2521–2528). The structures presented in DFT-minimized CoRE MOFs could also be presented in CoRE MOF database. The total structures with unique CCDC reference code is definitely less than 11558 (page10 line 188). Are there any other databases that were used in this work but not cited?

The reviewer is exactly correct here. The original CoRE MOF database is 5109 structures. The set of MOF structures in this work was a development version of CoRE MOF database version 1.0, where the porosity filter (e.g. pore limiting diameter > 2.4 Angstroms) was not applied. We apologize for the confusion and have added the phrase above in the manuscript.

We would like to add that initially, these structures with small pore limiting diameters were taken into consideration because we were hoping that porosity would open up with defects, but then quickly realized that this would be misleading as for non-porous MOFs, defects would need to be correlated heavily in such a way for the channels to open up and for the entire crystal to become porous. And as such, none of the candidates outside the original CoRE MOF set survived the criterion that we put forth, which is methane $K_H > 1 \text{e-}6 \text{ mol/kg/Pa}$. And to summarize, our results would have been exactly the same if we had used the original CoRE MOF database set as opposed to this unfiltered one.

Page 9: “A large-scale computational screening was conducted on an extended version of Computation-Ready Experimental (CoRE) MOF dataset² prior to application of the porosity filter”

2. The position of the removed linker should be clarified. In addition, when multiple linkers were removed, their relative position should also be indicated. If a missing linker defects present in the vicinities of inaccessible pore, the pore will be labeled accessible by the flood fill algorithm. The enhanced adsorption in Figure7 is strongly depending on the position of the linker vacant defects. Therefore, linker removal strategy should be explained in details.

We thank the reviewer for these comments. Similar concern has been brought up by Reviewer 1 and as such, we repeat our reply to him below:

In the manuscript, a single linker from the unit cell is being removed to observe any uptake enhancement, and this linker was chosen purely at random. For the candidate MOFs, high symmetry formed by the linkers and metal nodes leads to the same uptake enhancement being observed for all possible sites of linker vacancy within the unit cell. We have tested this for all of the 13 final candidates, where at least four different defect configurations that only differ in the relative location of the missing linker defect were created and tested for any differences in methane uptake enhancement. Results, presented below for each candidate MOF containing water & OH coordinating defects, indicate that there exists virtually no error or deviation in uptake between different positions of the linker defect within the unit cell.

None of the defect rates considered in the manuscript (highest being 8.333% or 1/12) allowed for more than one linker vacancy to be present within the unit cell. As a result, our study never directly considered the case of having multiple defects within a single unit cell.

Different defect rates reported in Figure 7 were achieved by expansion of the original unit cell to a 2x1x1 or 3x1x1 supercell and introduction of a single linker defect into those expanded cells. With the cell expansion, the number of secluded inaccessible pores within the calculation cell increases. Then, change in uptake enhancement is solely dependent on change in the proportion of inaccessible pores within the unit cell that would be opened up as a result of unit cell expansion. However, even with cell expansion, removal of any linker within the unit cell leads to uptake enhancement regardless of its position. The identity or relative location within the unit cell of the inaccessible pore being affected may change, but amount of uptake enhancement caused by a single linker vacancy remains equal for all linker positions. Thus, results of Figure 7 are still independent of the position of the linker vacancies.

Finally, Reviewer 1 brought up a similar point but there can be differences in the enhancement depending on the distributions of the linker defects. Most notably,

we can consider two scenarios, correlated vs randomized defects. Below, we state in verbatim our replies to Reviewer 1 with regards to this issue:

“The defect introduction scheme used in our manuscript can be considered as being “correlated”, in the sense that the linker vacancy was introduced to the unit cell, and this image was replicated infinitely in all 3 dimensions to express the defect crystal as a whole. This results in every single unit cell within the material experiencing the same uptake enhancement in a very ordered and correlated manner. However, this can be considered highly unrealistic, and the highly correlated defect distribution guarantees maximum uptake enhancement will be reached under the given defect proportion.

During the revision process, we newly considered the extreme case of purely randomized distribution of defects, meaning that the number of defects per unit cell would no longer be fixed and can hold any value between 0 up to total number of linkers per unit cell (under the constraint of fixed defect percentage in the entire crystal). This means that some unit cells can be defect-free without any uptake enhancement, whereas other unit cells can contain one or more defects leading to uptake enhancement. We believe this may be a more realistic way of considering the defect distribution into account. However, such randomness in material structure cannot be effectively taken into account in the GCMC simulations due to inevitably large supercell size that is required in including all representative proportions of defect scenarios within individual unit cells. As such, we assumed that the overall adsorption properties can be divided into appropriate linear combination of different unit cells (with different defect proportions) and calculated the uptake enhancement by simply considering the relative proportions of pristine and defect unit cells. The weights given to each of the defective unit cells (e.g. percentage of unit cells having 0, 1, 2, 3, ... defects) were based on how likely these were to be generated based on random removal of defects.

The two defect distribution schemes are shown above. A comparison of the two defect distribution schemes is made in the below bar graph, which shows the uptake enhancement of each of the 13 candidate MOFs as predicted by the correlated defects and randomly distributed defects. Proportion of linker vacancy used for calculation is different for each candidate due to the difference in the number of linker per unit cell. These are found in Table 1 of the manuscript.

The results show small differences between the two defect expression schemes for most of the candidate MOFs. This can be explained as follows: In our analysis,

we have kept the defect proportion relatively small (~ 2 to 8% of the total number of linkers). And as such, even in the “correlated” defective scheme, there still are many inaccessible regions that have not opened up due to the small proportion of linker defects. Consequently, the difference between correlated and random defective distribution, considering that not all the inaccessible regions were opened up in the first place, is not too big such that the overall message of the paper will be affected.

We would like to add that one key reason on why we kept the linker defect percentage to be relatively small in the first place was to not “oversell” our results. That is, surely with even higher defect percentages (e.g. 25%), we could have reported larger enhancement values. However, then there is an issue of framework stability that becomes more difficult to justify in the screening work, which might invite a whole level of skepticism towards the entire work. Moreover, it is conceivable that some of the readers (when they see a very high defective percentage value such as 25%) might think that it is not the inaccessible regions but the void space left by the linkers that is playing more of a crucial role in the enhancement. And we wanted to clearly deliver the message that this was not the case and that our mechanisms for enhancement was different.”

Page 16: “It is important to understand that the resulting distribution of linker vacancies as expressed by the above methodology is considered to be “correlated”. With the application of the periodic boundary condition, the newly created defective unit cell will be replicated infinitely in all three dimensions. Then, the exact same defect configuration will be used throughout the crystal, as shown in Figure S7. A more realistic scenario may be a purely random distribution of defects, where the number of defects per unit cell can vary, resulting in some unit cells with no defects and some with multiple defects. However, such random distribution of defects cannot be directly considered in GCMC simulations as it would at least require an immense unit cell that sufficiently considers all different defect scenarios and mitigates the effects of correlation. Thus, this study primarily reports the case of correlated distribution of defects, which can be directly tested with the current scheme of GCMC simulations. The case of purely random distribution of defects is still considered via indirect methods and is presented in the Supporting Information.”

Page 17: “Also, as can be seen from Figure S8 and Table S7, while there is small overall degradation in the enhancement across all of the MOFs, this enhancement trend still holds even for the random distribution of defects.”

Figure S8:

Table S7:

MOF CSD Refcode	CH ₄ uptake enhancement for each defect rate (cm ³ /cm ³)		
	8.33% defects	12.50% defects	25% defects
ABEMIF	27.457	35.944	49.254
AXUBOL	44.672	51.374	56.163
HOMZEP	31.619	43.440	62.876
JEWYAM	29.613	38.766	53.121
KOCWEF	36.374	46.547	58.488
MUWQEB	22.400	29.324	40.183
PAMHIW	40.208	49.564	60.099
QAGQEW	30.661	40.297	52.204
REGYOT	33.476	44.206	57.571
UTEWOG	34.711	45.440	62.266
UTEWUM	28.119	36.811	50.442
VEXYON	24.345	31.869	43.671
XENZUN	32.472	42.509	58.251

SI: Section 9 (SI pg. 14-17)

3. It would be more relevant to simulate the methane uptake of the defected MOFs by removing the terminal waters, as the terminal water on most 2+ metals could be removed during the activation of MOFs.

This is an excellent point made by the reviewer and below, we show the data for 4 of the tested cases where coordinated water molecules from defect expression is removed after relaxation:

In all four cases, removing the water does not effectively change the results of the simulation. We expect most of the other candidate MOFs and their defect scenarios to show similar behavior and as such, we did not go through all the rest of the candidate materials.

There are several reasons as to why this is the case: 1) our linker vacancy schemes involve a very smaller proportion of water molecules due to the small proportions of linker being removed. 2) Interaction between methane and MOF is relatively weak due to the absence of electrostatic interactions. 3) In almost all cases, the newly created opening into the previously inaccessible pores is already large when defect sites are coordinated with hydroxyl and water. This makes the uptake enhancement effect highly independent from the presence/absence of the terminal water groups. However, there does exist two exceptions to this third claim, and these are extensively considered below.

Reviewer's comment led us to draw our attention to the only two exceptions in which we were unable to observe any uptake enhancement even when the smaller water and OH groups were used for coordination: IN linker vacancy in AXUBOL, and linker 1 vacancy in KOCWEF. For these MOFs, when replacement of the linker with hydroxyl and water is performed, the newly created opening into the inaccessible pores were still too narrow for methane to enter the inaccessible pores. We wanted to see how this would change when the terminal waters are removed, as suggested by the reviewer. Results, shown below, showed that removal of terminal water at the defect sites had finally resulted in the connection between main channel and the unit cell, and uptake enhancement was now being observed for these scenarios. As we agree with reviewer's point that it is more relevant to consider the case where MOF is fully activated, we have updated the methane uptake results with that of the terminal-water removed case for these two defect scenarios.

We would also point out that removing water for the defective MOFs can lead to the creation of new open metal sites, but we still opted to use UFF in this case. We have included below a figure of our methane simulation data using TraPPE/UFF in M-MOF-74 (an open metal site MOF structure) and experimental data for M = Co, Mg, Mn, Ni, and Zn (T = 298 K). As can be seen, the generic

UFF parameters does a good job of describing methane-MOF interaction at the Henry regime and thus we did not find it necessary to go through deriving force fields for all the open metal sites that might occur due to washing away of water.

Following changes have been made with regards to the water removal issue:

Page 15: Only in a few cases when no methane uptake enhancement was initially observed, the terminal water molecules were removed as an extra measure since they can be deemed extraneous to overall coordination state of the metal clusters (see SI).

SI: Section 7 (SI pg. 11-12)

4. ABEMIF, UTEWOG, VEXYON, and XENZUN are anionic frameworks with metal cations/[$(\text{CH}_3\text{NH}_2)^+$] in the cavity. The cations could affect the gas adsorption, however, they are omitted in the CoRE MOF database. Are they considered in the simulation?

This is an important point that we initially looked over. We did a careful reading of the original manuscripts and found that among the 13 final candidate materials, 6 were anionic MOFs that contained variety of different cations. They are JEWYAM, MUWQEB, XENZUN, VEXYON, ABEMIF, and REGYOT.

One thing in common about these MOFs is that the cations can be exchanged, which allows for the bulkier organic cations such as $[(\text{CH}_3)\text{NH}_2]^+$ to be replaced with a much smaller alternative like the metal cations. The original publications report the exchange of ionic complexes/organic cations for metal cations (in parentheses): JEWYAM (Mn), MUWQEB (Fe), XENZUN (Cu), VEXYON (Cu), and REGYOT (Li). Since having these smaller cations would take up much less volume and minimize the effect on methane adsorption properties, we decided to achieve charge neutrality on these MOFs with their respective metal cations. In the case of ABEMIF, metal cation was never used in the original publication. However, noting that ABEMIF and REGYOT shares the identical metal cluster, we decided to also consider ABEMIF in the same manner as REGYOT, where cation exchange with Li^+ would have been performed on the MOF.

We were lucky to find that extra-framework metal cation positions were reported in literature for JEWYAM, MUWQEB, and VEXYON. For these MOFs, we manually located the metal cations in very close proximity to what was shown in the original literature and conducted an additional MOPAC relaxation to stabilize the cations within the anionic frameworks.

In the case of REGYOT, we had actually taken the presence of cations into account before, as we had to reproduce the experimental data with cations included. Although the cation position is never explicitly shared, we deduced their position by the following course of logic:

We first noted that original publication reports significant methane uptake difference in the high pressure regime for the framework with different cations, $[\text{Et}_2\text{NH}_2]^+$ and Li^+ . Considering that methane adsorption behavior in the high pressure regime is highly volume dependent, we thought this is most likely caused by the presence of differently sized cations in the main channels. Then, recognizing that the charge balance is more or less needed around the metal cluster, we then located the Li^+ cations nearly the metal clusters within the main channel, and performed MOPAC relaxations to stabilize the cation positions. This course of logic led us to closely match the adsorption profile REGYOT with Li^+ as presented in Figure 8(b).

For ABEMIF, given that it is identical to REGYOT in its metal cluster type and coordination environment, we performed the exact same procedure on ABEMIF with Li^+ to obtain its charge neutralized system.

For XENZUN, no information was explicitly given regarding the location of metal cations. However, the original publication denotes that the cations are present

within the main channels rather than the cage, so we adopted a similar approach as REGYOT to estimate the positions of Cu cations within the framework with MOPAC.

This cation insertion and stabilization process was conducted also for all defect scenarios. Below, we present how the methane adsorption isotherms have changed for each anionic MOF:

Since charge neutrality was achieved with the smaller metal cations, there exists virtually no difference from the previous results for most of the MOFs. Observable difference is found for ABEMIF and XENZUN, which are significantly smaller in structure size compared to other MOFs. As this is a more accurate representation of each anionic MOF, the methane adsorption data for the anionic MOFs in the manuscript have been updated with the new data after charge neutrality has been achieved. The cation insertion and stabilization process is also discussed in the SI.

Note that in considering the location of the counter-cations during the GCMC simulations, the metal cation was fixed in the energy minimized position from MOPAC relaxation. We validate this fixed-cation assumption with the following reasons: 1) fixed-cation method for REYGOT yields good agreement between computational and experimental data. 2) number of cations is sufficiently small, and thus it cannot make too much of a difference on adsorption properties. 3)

The cation should stay near the metal cluster due to strong interactions, as the charge balance is required around the metal coordination site.

Again, we sincerely thank the reviewer for pointing this out.

Page 13: "In the process of MOPAC relaxation, it was found that the candidate pool contained several anionic MOFs whose counter-cations were missing in the structure files. The cations were then manually added for the correct representation of these MOFs, and the cation insertion procedure is briefed case by case in the Supporting Information."

SI: Section 5 (SI pg. 9)

Reviewer #3 (Remarks to the Author):

In this paper, Chong et al. report a new strategy for improving existing MOFs for gas sorption: finding frameworks which contain inaccessible void volumes that could be connected to the accessible volume by introducing ligand vacancies. They then carry out a screening on tens of thousands of reported MOF crystal structures and find that significant improvement in CH₄ uptake could be achieved in 13 of these structures. The authors report an ingenious strategy, and follow it through to its logical theoretical conclusion, yielding suggestions that could be directly experimentally tested.

I have a few technical questions:

-the authors refer to an energy of kT throughout: what T are they using?

We are using $T = 298$ K as this is the relevant temperature for all of our GCMC uptake simulations. We apologize for this lack of clarification and have added the following sentence in the manuscript.

Page 8: "In identifying the low energy regions for adsorption, a previously used energy threshold of $15 k_B T$ (with $T = 298$ K) was utilized,³⁶"

-the authors relax the structures using MOPAC to check for structural stability post defect inclusion, and find in one case that the structure changes by about 15%V. Did the authors also relax the non-defective structure in MOPAC to check that the changes are not due to the different simulation protocol?

This is a very good point brought upon by the reviewer. Yes, we did also relax the non-defective structures. The volume difference was previously observed for the MOF HOMZEP, where the volume changes from non-defective to defective was around 15%.

Additional analysis during our revision process revealed that some of the oxygen atoms expressed in the structure files of the CoRE MOF dataset are actually supposed to be hydroxyl groups, but they were lacking hydrogen atoms. This was the case for HOMZEP and also PEYVEV, which is a topological twin. We suspected that the severe distortion of HOMZEP during the relaxation process was caused by this issue, and as such with proper substitutions of the atoms, we found that the volume change was reduced to 0.03% and 0.98% for HOMZEP.

Strangely, however, proper expression of the coordination state in PEYVEV has led to an unusual linker detachment even in the pristine framework. We suspect this to be shortcomings of using semi-empirical methods for certain metal types. Thus, HOMZEP now replaces PEYVEV in the list of final candidates. Relevant sections regarding the volume change of HOMZEP has been removed from the main manuscript, and PEYVEV is removed from the candidate pool due to an abnormal linker detachment during PM7 relaxation.

Following clarification is added in the manuscript regarding the removal of PEYVEV, and all parts related to previous issues with HOMZEP have also been removed or edited:

Page 13: “Two other MOFs (EZOFEF, PEYVEV)^{54,55} were ruled out for showing significant structural collapse or linker detachment with MOPAC energy minimization”

Page 16: “Results, presented in Table S6, show that all of the MOFs show no significant framework collapse with linker vacancy defects.” (No more discussion of HOMZEP collapse)

The text is in general clear and explains the ideas well. Unfortunately, the authors' extremely diverse vocabulary and elaborate style sometimes impeded my ability to understand their intended meaning: for example, is a 'volume offset' the percentage change in volume or in lines 101-103, do the authors believe that only UiO-type MOFs could be affected by defects, or do they think that other people could come to this conclusion? If the authors were to go through and refocus their text to simplify some of the writing I think it would greatly help readers.

We apologize for being verbose. With regards to “volume offset”, we believe that this is a confusing term as we meant to say, “volume difference” (i.e. % changes in the volume pre and post relaxation of the MOF structures). As such, we have clarified these issues. Moreover, we went through the manuscript and made following changes to make our messages clearer. Again, we thank the reviewer.

With regards to UiO-66 type of MOFs, it is our belief that most of the community is currently investigating this structure (or its variants) with regards to defects in MOFs and we believe that there are other MOFs (like the ones that we presented in this manuscript) that can lead to interesting property transformation with small amounts of defects.

Page 16: “We presumed a resulting volume difference of less than 5% is sufficient for assuring the feasibility of introducing linker vacancies in the current stage of our research”

Two other aspects of the manuscript are not ideal for readers. A few times I encountered abbreviations and symbols which were not yet defined (BDC, IN, XSHV, KH): in some cases I had to consult the SI, and in one or two, I am still unclear as to their meaning (XSHV?).

Again, we apologize for the inconvenience. For the words “BDC”, “IN”, “KH”, “XSHV(Z)”, these are the respective meanings: “BDC” = 1,4-benzenedicarboxylate (name of a MOF ligand); “IN” = isonicotinate (name of a MOF ligand); “KH” = “ K_H ” = symbol for Henry’s constant; “XSHZ” = *N*-acylsalicylhydrazide.

We have clarified these issues by providing meanings behind these abbreviations when they are first used. See below for changes made in the manuscript/Supporting Information.

Page 9: “CH₄ Henry coefficients (KH) were calculated for each MOF twice”

Additionally, although I greatly appreciated the inclusion of diagrams of the crystal structures in the SI, the perspectives sometimes chosen are sometimes not very informative (e.g. AXUBOL on page 10). I would also very much like if the authors presented their final structures: could they show diagrams of the structures after they have substituted in the defects, and also include CIFs or similar of the structures? At present, figure 5 and 9 are the only figures to give some structural information about the resultant structures.

We agree that this is an important point. As such, in addition to the pre-existing diagrams, we have included the actual structure files for all the top structures in the SI.

REVIEWERS' COMMENTS:

Reviewer #1 (Remarks to the Author):

I believe the authors responded to all comments and improved the manuscript significantly in the review process. I recommend publication.

Reviewer #2 (Remarks to the Author):

In my previous review for this manuscript, I have stated the importance of this paper, its novelty, and good level of discussion. The questions raised in the previous review were properly addressed in the revised manuscript. I am therefore supportive of acceptance of the paper in its present form.

REVIEWERS' COMMENTS:

Reviewer #1 (Remarks to the Author):

I believe the authors responded to all comments and improved the manuscript significantly in the review process. I recommend publication.

We thank Reviewer 1 for their time and the valuable comments made throughout the revision process.

Reviewer #2 (Remarks to the Author):

In my previous review for this manuscript, I have stated the importance of this paper, its novelty, and good level of discussion. The questions raised in the previous review were properly addressed in the revised manuscript. I am therefore supportive of acceptance of the paper in its present form.

We thank Reviewer 2 for their time and the valuable comments made throughout the revision process.